

# Comparative transcriptome analysis on candidate genes associated with fruiting body growth and development in *Lyophyllum decastes*

Shanwen Ke[1,2,*], LingQiang Ding[1,2,*], Xin Niu[1,2], Huajia Shan[1,2], Liru Song[1,2], Yali Xi[1,2], Jiuhai Feng[1,2], Shenglong Wei[1,2] and Qianqian Liang[1,2]

[1] Gansu Engineering Laboratory of Applied Mycology, Hexi University, Zhangye, Gansu, China
[2] Gansu Key Laboratory of Genetics and Breeding of Edible Fungi, Hexi University, Zhangye, Gansu, China
* These authors contributed equally to this work.

Corresponding author
Qianqian Liang, qql1128@126.com

## ABSTRACT

*Lyophyllum decastes* is a mushroom that is highly regarded for its culinary and medicinal properties. Its delectable taste and texture make it a popular choice for consumption. To gain a deeper understanding of the molecular mechanisms involved in the development of the fruiting body of *L. decastes*, we used RNA sequencing to conduct a comparative transcriptome analysis. The analysis encompassed various developmental stages, including the vegetative mycelium, primordial initiation, young fruiting body, medium-size fruiting body, and mature fruiting body stages. A range of 40.1 to 60.6 million clean reads were obtained, and *de novo* assembly generated 15,451 unigenes with an average length of 1,462.68 bp. Functional annotation of transcriptomes matched 76.84% of the unigenes to known proteins available in at least one database. The gene expression analysis revealed a significant number of differentially expressed genes (DEGs) between each stage. These genes were annotated and subjected to Gene Ontology and Kyoto Encyclopedia of Genes and Genomes analyses. Highly differentially expressed unigenes were also identified, including those that encode extracellular enzymes, transcription factors, and signaling pathways. The accuracy of the RNA-Seq and DEG analyses was validated using quantitative PCR. Enzyme activity analysis experiments demonstrated that the extracellular enzymes exhibited significant differences across different developmental stages. This study provides valuable insights into the molecular mechanisms that underlie the development of the fruiting body in *L. decastes*.

## INTRODUCTION

Mushroom-forming fungi are widely distributed in nature and play an important role in conversion of low quality agricultural waste, such as plant straws, sawdust, cotton stalks,

and cottonseed skins, into nutrient-rich fruiting bodies (*Liu et al., 2022*). Mushrooms are highly valued for their delicious taste, nutritional benefits, and medicinal properties. In the case of basidiomycetes, the fruiting body is a complex multicellular structure and organic whole. So, studying fruiting body development is crucial to understanding the mechanisms underlying the transition from simple to complex multicellular fungal structures. The process of fruiting body initiation and development is a complex and rapid event in the mushroom life cycle that is influenced by cellular processes, genetics, physiology, and environmental factors (*Wessels, 1993*). The molecular mechanisms underlying mushroom fruiting body formation and development have become a prominent area of research in mycology, with significant implications for mushroom yield and quality. Recent advances in molecular biology have led to the identification and study of key genes involved in fruiting body formation and development (*Tao et al., 2013*), such as glycoside hydrolase family 55 exo-β-1,3-glucanase-encoding genes in *Volvariella volvacea* and 11 *Flammulina velutipes* laccase genes (*Wang et al., 2015*). These studies have shed light on the complex genetic and biochemical processes involved in mushroom fruiting body formation and development. A jacalin-related lectin gene from *F. velutipes* was found to be involved in the growth of aerial mycelium and fruiting body formation (*Lu et al., 2016*). In basidiomycetes, mushroom development starts with branching vegetative mycelium that further differentiate forming the primordium which contains all the tissues observed in the mature fruiting body (*Kües & Liu, 2000*). According to *Song, Kim & Kim (2018)*, the mushroom fruiting body formation was regulated by specific environmental stimuli, and subsequently regulated by signal transduction pathways, which results in morphological changes during development. This process involves stage-specific expression in order to produce beneficial properties.

The gray basidiomycete *Lyophyllum decastes*, belonging to the Tricholomataceae family, is a valuable edible and medicinal mushroom renowned for its delectable taste, superior texture, rich nutrients, and potent medicinal benefits (*Nakamura et al., 2007*; *Pokhrel & Ohga, 2007*; *Zhang et al., 2022*). Polysaccharides, the main component of the *L. decastes* fruiting body, has been used as a traditional medicine with specific medicinal properties for the treatment of cancer and diabetes, as well as lowering blood lipids (*Ukawa et al., 2002*). The widespread availability and utilization of *L. decastes* as a supplement for promoting health and preventing/treating various ailments has led to its considerable popularity. Its agricultural, commercial, and medical value make it an attractive option for consumers worldwide, with significant potential for growth and economic benefit in the fields of industry, medicine, and agriculture (*Ukawa et al., 2007*). Previous research on *L. decastes* mainly focused on disease prevention, genetics and bioactive substances (*Pokhrel & Ohga, 2007*; *Sunagawa et al., 2008*; *Parada et al., 2011*; *Clericuzio et al., 2013*). In spite of the critical role played by the fully developed fruiting body of *L. decastes* in commercial and medicinal applications, there exists a notable paucity of information concerning the molecular mechanism that governs its growth and development. Moreover, the precise genes pivotal to fruiting body development have yet to be fully elucidated.

The utilization of Illumina RNA sequencing technology for high-throughput transcriptome sequencing has enabled the accurate identification of mRNA transcript

abundance information, thereby facilitating research in the life sciences and significantly enhancing the efficiency of functional gene discovery in basidiomycota mushrooms (*Zhang et al., 2015b*; *Gao et al., 2023*). Several important genes regulating fruiting body development were identified by using comparative transcriptomic analysis during the mycelia, primordia, and fruiting body stages of *Pleurotus tuoliensis* (*Fu et al., 2017*). Similarly, the transcriptomes of *Lentinula edodes* were studied from the early bud stage to the fully developed stage of fruiting body growth, revealing the activation of specific metabolic pathways in the fruiting body (*Wang, Zeng & Liu, 2018*). Furthermore, the transcriptomes of the mycelium and fruiting body were analyzed in *Agrocybe aegerita* (*Wang et al., 2013*), *Schizophyllum commune* (*Plaza et al., 2014*), *Hypsizygus marmoreus* (*Zhang et al., 2015a*), and *Ophiocordyceps sinensis* (*Tong et al., 2020*), providing vast amounts of information about mushroom formation and some of the molecular mechanisms that control fruiting body development.

A comparative transcriptome analysis was conducted to gain a deeper understanding of the molecular mechanisms responsible for the development of mushroom fruit bodies in *L. edodes*. Through the utilization of high-throughput Illumina RNA-sequencing technology, 50 million reads were generated per sample, which allowed for the identification of 76,752 unigenes through *de novo* assembly. The analysis of differentially expressed genes (DEGs) revealed the critical role of genes and pathways in the fruiting process of *L. edodes*. The focus of our study was on candidate extracellular enzyme genes, including those that encode cellulase, galactosidase, and laccase, as these enzymes are vital for the growth and development of fruiting bodies. The transcriptome data obtained from this study will significantly enhance our understanding of the genetic and molecular mechanisms involved in the development of fruiting bodies in *L. edodes*. Furthermore, this data will provide valuable targets for molecular breeding aiming to improve productivity and quality and future research endeavors focused on *L. edodes*.

## MATERIALS AND METHODS

### Sample preparation

The *L. decastes* strain was preserved at the China General Microbiological Culture Collection Center. The mycelia were cultured on potato dextrose agar with a pH of 4–5 at 25 °C in the dark. The cultivation substrates, contained in plastic bags, were comprised of 750 g medium consisting of 75% cottonseed hull, 12% sorghum seed, 12% wheat bran, 0.5% sugar, and 0.5% superphosphate. The bags were incubated in a 19 °C greenhouse under artificial light conditions once fully colonized. *L. decastes* samples were collected from randomly selected bags at five different developmental stages: the vegetative mycelium stage (VM), primordium initiation stage (PI), young fruiting body stage (YF), medium-size fruiting body stage (MSF), and mature fruiting body stage (MF) (Fig. 1). The morphological changes were observed, which the VM turned into PI in 7–10 days, and following young fruiting body appeared (YF) in 3–5 days, and the medium-size fruiting body appeared (MSF) in 5–7 days, and finally fruiting body matured (MF) in 5–7 days. All samples were promptly cryopreserved in liquid nitrogen and stored at −80 °C for RNA extraction.
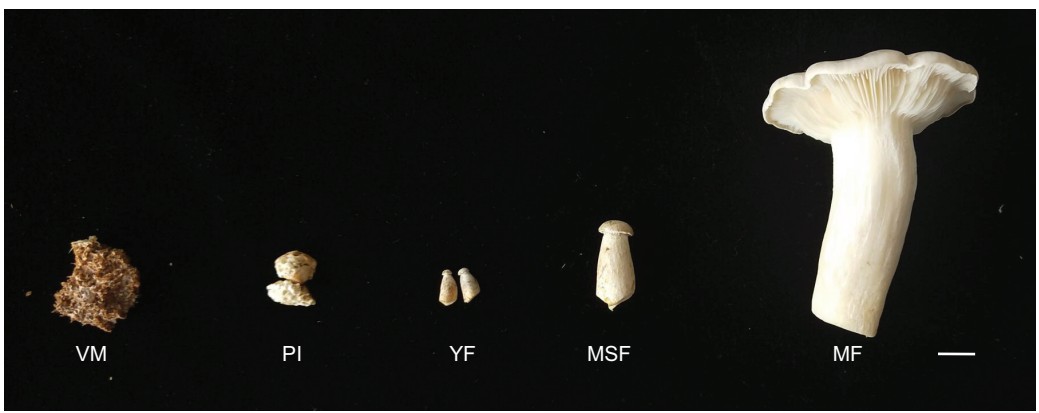

**Figure 1** *L. decastes* **samples were collected from randomly selected bags at five developmental periods.** The different developmental stages of *L. decastes*. VM, vegetative mycelium stage; PI, primordial initiation stage; YF, young fruiting body stage; MSF, medium-size fruiting body stage; MF, mature fruiting body stage; bar = 1.0 cm.

## RNA extraction and sequencing

The acquisition of total RNA from each sample was executed through the utilization of diverse kits, including the TRIzol Reagent (Invitrogen, Waltham, MA, USA). After extracting the RNA, its quality and quantity were assessed using an Agilent 2100 Bioanalyzer (Agilent Technologies, Palo Alto, CA, USA), NanoDrop (Thermo Fisher Scientific Inc. Waltham, Ma, USA), and 1% agarose gel. For each sample, 1 μg of total RNA with a RNA integrity (RIN) value greater than 7 was utilized for subsequent library preparation. The NEBNext First Strand Synthesis Reaction Buffer and NEBNext Random Primers were used for mRNA fragmentation and priming. ProtoScript II Reverse Transcriptase was utilized to produce the initial strand cDNA, followed by the Second Strand Synthesis Enzyme Mix to generate the complementary second strand cDNA. The size selection of Adaptor-ligated DNA was carried out using AxyPrep Mag PCR Clean-up (Axygen, Union City, CA, USA) and fragments measuring approximately 360 bp (with an approximate insert size of 300 bp) were recovered. The manufacturer's protocol (NEBNext® Ultra™ RNA Library Prep Kit for Illumina®) was followed in order to construct next-generation sequencing library preparations. The Illumina HiSeq machine was used according to the manufacturer's instructions (Illumina, San Diego, CA, USA) to load and multiplex the libraries with distinct indices. The statistical power of this experimental design calculated in RNASeqPower was 0.87. Three biological replicates were taken for each stage.

## Assembly and functional annotation

The initial paired-end reads underwent a series of quality control measures, which involved the elimination of adapter, low-quality, and duplicate sequences. The quality of the reads was evaluated by FastQC software (Version 0.10.1). In order to remove technical sequences, including adapters, polymerase chain reaction (PCR) primers, or fragments thereof, and quality of bases lower than 20, pass filter data of fastq format were processed by Cutadapt (Version 1.9.1, phred cutoff: 20, error rate: 0.1, adapter overlap: 1 bp, min.

length: 75, proportion of N: 0.1) to be high quality clean data. Subsequently, a *de novo* assembly was conducted using Trinity (version 2.4.0) (*Grabherr et al., 2011*), a novel approach that facilitated the efficient and robust reconstruction of transcriptomes from RNA-seq data. This approach employed three independent software modules, namely Inchworm, Chrysalis, and Butterfly, in a sequential manner to process large quantities of RNA-seq reads. The contigs obtained were further refined using Cd-hit (version 4.6.2) to eliminate duplicates, resulting in the production of the unigene sequence file. To annotate the assembled unigene sequences, they were searched against several databases, including non-redundant protein of NCBI (Nr, https://www.ncbi.nlm.nih.gov/protein), Gene Ontology (GO, http://www.geneontology.org), Cluster of Orthologous Groups (COG, http://www.ncbi.nlm.nih.gov/COG/), Kyoto Encyclopedia of Genes and Genomes (KEGG, http://www.kegg.jp/), and Swiss-Prot (http://www.expasy.org), using blast software (version 2.2.28+) with an E-value of less than $<10^{-5}$.

## Expression and enrichment analysis

The reference gene file utilized in this study was the unigene sequence file. Using RSEM (v1.2.6), gene and isoform expression levels were estimated from the pair-end clean data, according to the methodology proposed by *Fu et al. (2012)*. In order to identify the genes that were differentially expressed, the DESeq2 (version 1.26.0) Bioconductor package was utilized (*Anders, McCarthy & Chen, 2013*; *Love, Huber & Anders, 2014*), which employed a model based on the negative binomial distribution. The adjustment for controlling the false discovery rate (FDR) was carried out using the Benjamini and Hochberg approach, the detection of DEGs was set at a significance level of FDR < 0.05 and |$\log_2$ fold change| > 1. The genes that were enriched were determined using GO-TermFinder (version 0.86), a tool that assigned annotations to a set of genes based on a statistically significant *p*-value less or equal than 0.05. Additionally, we developed in-house scripts to identify significant DEGs in KEGG pathways. Pathway significance enrichment analysis using KEGG pathway as unit, hypergeometric test was applied to find out the pathway of significant enrichment in DEGs compared with the reference background.

## qPCR

To ensure the accuracy of the RNA-Seq data, a representative selection of DEGs was chosen. These genes were involved in the development of *L. decastes* fruiting bodies and encoded cellulose, galactosidase, and laccase. The fragments per kilobase of exon model mapped fragments (FPKM) values for these genes are available (Table S8), while the primer sequences are listed in Table S9. RNA extraction was carried out using RNAios Plus manufactured by Takara, and cDNA was synthesized using the Revert Aid First Strand cDNA Synthesis Kit provided by Thermo Scientific. The transcriptome data was utilized to obtain specific primer pairs, which were subsequently employed for qRT-PCR. The CFX96 TouchTM Real-Time PCR Detection System (Bio-Rad, Hercules, CA, USA) was used to conduct the qRT-PCR in triplicate. The reaction mixture consisted of 20 μL, which included 10 μL of 2×SYBR Green mixture (TaKaRa, Dalian, China), 1 μL of diluted cDNA, and 1 μL each of specific forward and reverse primer. The $2^{-\Delta\Delta CT}$ method was used to

analyze the relative gene expression, with β-Actin serving as an internal control (*Deepak, Ishii & Park, 2008*).

## Determination of extracellular enzyme activity

The culture medium was sampled at the VM, PI, YF, MSF and MF stages of *L. decastes* in order to determine extracellular enzyme activity. The activities of cellulase, galactosidase, laccase and peroxidase were determined by Lanzhou Boruke Biotechnology Co., LTD., China.

## RESULTS

### Illumina sequencing and assembly

The objective of this study was to identify changes in the transcriptome of *L. decastes* at five distinct developmental stages. To achieve this, RNA samples were obtained from the VM, PI, YF, MSF, and MF stages, which were then subjected to high-throughput sequencing. Overall, approximately 63 Gbp of raw data were produced from 15 samples across the VM, PI, YF, MSF and MF stages (Table S1). After data filtering and trimming, a total of 743,342,738 high-quality clean reads were generated with a Q20 percentage >97.3% (Table S2). Utilizing Trinity software, the reads were assembled into 54,338,739 contigs with a mean length of 86.2 bp and an N50 length of 103 bp (Table S3). The contigs were initially assembled, and a further assembly process was subsequently conducted, resulting in the formation of 15,451 unigenes. According to a recent research, 14,499 gene models have been predicted from the genome of *L. decastes* (*Xu et al., 2023*). Compared with these previously predicted genes, the average GC content of the unigenes in this research (51.78%) is similiar to that of *L. decastes* LGR-d1-1 (50.25%) and LRG-d1-5 (50.38%). Moreover, we *de novo* assembled some putative novel unigenes, and will provide important information for further annotation and characterization of *L. decastes*. The length of these unigenes ranged from 201 bp to 24,176 bp, with an average length of 1,462.68 bp and N50 length of 2,921 bp (Table S3). Of all the unigenes, 6,679 (43.23%) were between 200 to 1,000 bp long, 6,280 (40.64%) were within the range of 1,000~3,000 bp in length, while 16.13% were longer than 3,000 bp. Notably, none of the unigenes were shorter than 200 bp (Fig. 2). In order to enhance the comprehensiveness of the *L. decastes* transcriptome sequencing, the assembled unigenes were utilized to map the clean reads. The mapping outcomes are presented in Table S2.

### Sequence annotation

The compiled unigenes were subjected to annotation through the utilization of the BLASTx algorithm against sequences sourced from four publicly available gene databases. A total of 11,783 (76.84%) unigenes were annotated in at least one database, 10,907 (70.59%) unigenes were observed to matched known proteins in the Nr database (E-value < $1.0E^{-45}$), followed by the KEGG (6,650, 43.04%), SwissProt (6,454, 41.77%) and COG (5,324, 34.46%) database (Table S4 and Fig. S1). However, only 912 (5.90%) unigenes matched in all database. After functional annotation, the species distribution analyses of the annotated unigenes were calculated from the annotation results. We found that the top

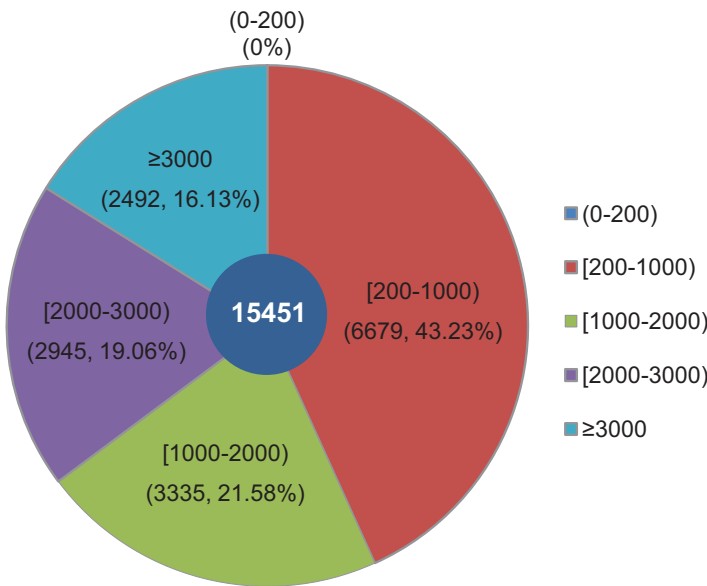

**Figure 2** **Among all unigenes, 6,679 (43.23%) had lengths ranging from 200 to 1,000 bp, 6,280 (40.64%) were within the range of 1,000~3,000 bp in length, 16.13% were more than 3,000 bp, and none were less than 200 bp in length.** The length distribution of assembled unigenes from *L. decastes*. Different colors represent different lengths of unigenes.

ten representative annotated species in Nr database were *L. decastes* LRG-d1-1 (66.13%), *H. marmoreus* (25.45%), *Termitomyces* sp. J132 (7.98%), *Galerina marginata* CBS 339.88 (1.93%), *Laccaria amethystine* LaAM-08-1 (0.95%), *Pelomonas* sp. Root 1217 (0.84%), *Hebeloma cylindrosporum* h7 (0.78%), *Laccaria bicolor* S238N-H82 (0.69%), *Piloderma croceum* F1598 (0.65%) and *Pelomonas* sp. Root1444 (0.63%), as shown in Fig. 3. Significantly, BLAST comparison was performed between the 15,451 unigenes in this study and 14,499 genes in the study by *Xu et al. (2023)*, which revealed that 74.25% unigenes can be found in the 14,499 genes with an identity of 85%.

In order to evaluate the comprehensiveness of the transcriptome and the effectiveness of the annotation process, an investigation was carried out on the annotated unigenes involved in GO classifications. The three primary GO categories, namely biological process, cellular component, and molecular function (Fig. S2), made up a total of 4,082 unigenes that were classified into 59 functional groups. The matched unigenes were allocated to 25 classifications by the COG database, with the most extensive category being exclusively related to general function prediction and encompassing 5,324 unigenes (Fig. S3). Moreover, the unigenes that underwent annotation in the KEGG database were categorized into six distinct biological pathways, which include cellular process, environmental information processing, genetic information processing, human diseases, metabolism, and organismal system (Fig. S4).

## Analysis of differentially expressed genes (DEGs)

To identify and evaluate the transcriptional profiles during *L. decastes* fruiting body formation and development, the expression levels of unigenes from five developmental

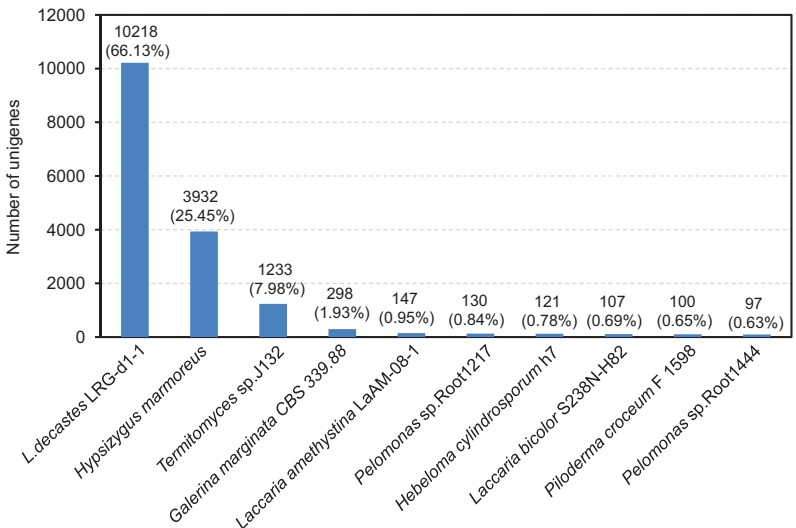

**Figure 3 After functional annotation, the species distribution analyses of annotated unigenes were calculated from the annotation results.** The nine most represented species among the annotation results of the *L. decastes* unigenes. Species distribution of BLASTx results against the NCBI non-redundant protein database (E-value < 1e$^{-10}$) and the proportions for each species are shown.

stages were calculated using the Bioconductor software package and DESeq2 (*Anders, McCarthy & Chen, 2013*; *Love, Huber & Anders, 2014*), setting a FDR value ≤ 0.05 and log$_2$ ratio ≥ 1. Pairwise comparisons showed that 2,502, 713, 208 and 1,449 unigenes were significantly differentially expressed between PI and VM, YF and PI, MSF and YF, and MF and MSF, respectively, and 1,568, 1,673 and 2,704 unigenes were significantly differentially expressed between MF and YF, MF and PI, and MF and VM, respectively (Figs. 4, S5, and Table S5). As shown in Fig. 4, most DEGs occurred between the MF and VM (1,278 up-regulated, 1,426 down-regulated) and PI and VM (1,185 up-regulated, 1,317 down-regulated), and only a minimal number of unigenes were identified as DEGs between MSF and YF (70 up-regulated, 138 down-regulated), illustrating that a significant number of genes were responsible for the conversion of mycelium into fruiting body. When compared to the initial phase of fruiting body development, a higher quantity of genes displaying significant alterations was detected during the subsequent stage. This suggested that the maturation period of the fruiting body is a complex and multifarious process. As predicted, there was a gradual increase in the number of differentially expressed genes (DEGs) during the transition from mycelium to a fully matured fruiting body. Notably, in all compared groups, the quantity of unigenes that were down-regulated surpassed the quantity of those that were up-regulated. To investigate gene expression relationships, Venn diagrams of different pairwise comparisons were introduced to show the statistics of the DEGs in depth (Fig. 5). In total, only 19 unigenes were differentially expressed both in PI *vs* VM, YF *vs* PI, MSF *vs* YF and MF *vs* MSF. However, 323 DEGs were detected in MF *vs* MSF, MF *vs* YF, MF *vs* PI, and MF *vs* VM, simultaneously. These genes, which were significantly differentially expressed at each stage, may play a crucial role in the development of the *L. decastes* fruiting body.

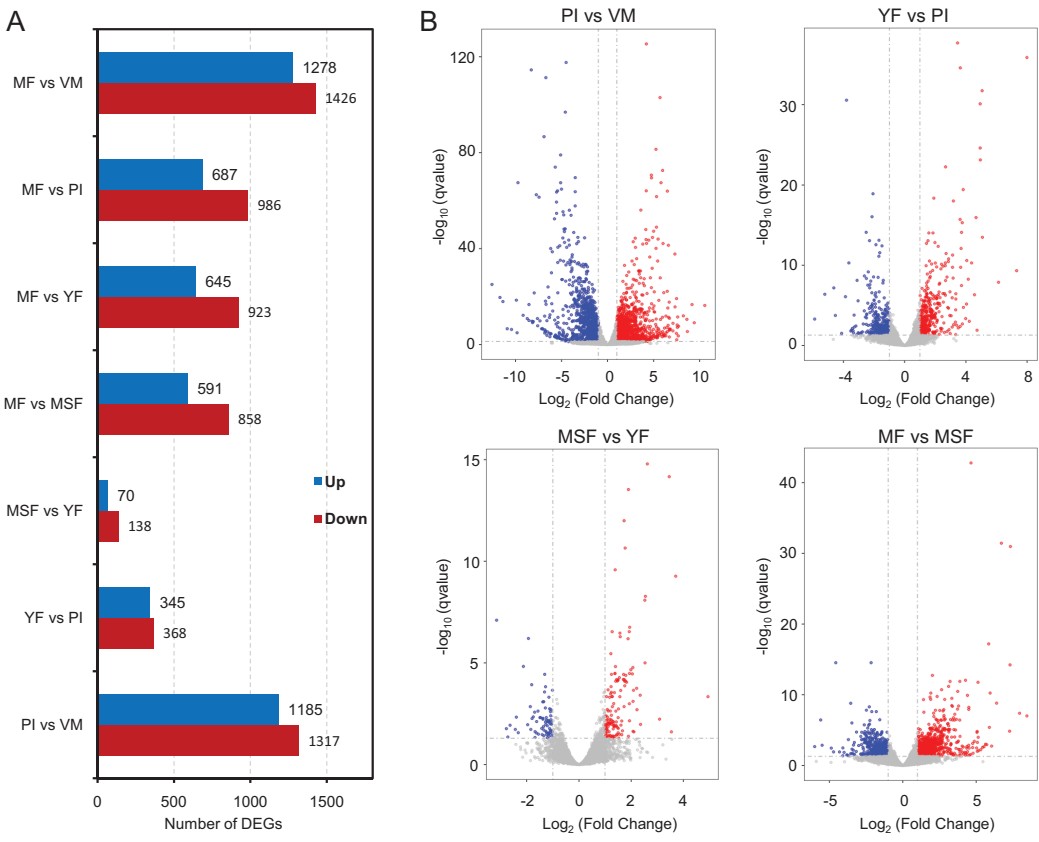

**Figure 4 Analysis of differentially expressed genes between different development stages in *L. decastes*.** Summary of the significant DEGs between mycelium, primordium and fruiting bodies of *L. decastes*. (A) The number distribution of significantly up-regulated and down-regulated unigenes in each comparison is summarized. (B) Volcano plots of DEGs across different developmental stages. The blue dots represent up-regulated genes, the red dots represent down-regulated genes.

Some main up-expressed and down-expressed DEGs were detected between different development stages of *L. decastes* (Table 1). We found that the expression difference of these genes in PI *vs* VM were significantly higher than those in other comparison groups. Of these unigenes, with log$_2$ fold change values ranging from −9.42 to 10.06 some were regarded as new genes that encoded unknown proteins or hypothetical proteins that could not be matched in any database. In different comparisons, some unigenes showed different up-regulation or down-regulation and need to be further studied.

## Gene ontology (GO) enrichment analysis of DEGs

With the objective of identifying the potential contributions of genes that were differentially expressed, we carried out a GO functional enrichment analysis to assign their functions to the process of fruiting body development. Subsequently, the DEGs were subjected to GO term enrichment analysis, which enabled the characterization of the relevant GO terms. The results showed that the DEGs exhibited connections with a multitude of biologically significant activities. The DEGs were sorted into three main GO categories, with each consisting of 30 distinct functional subcategories. The number of

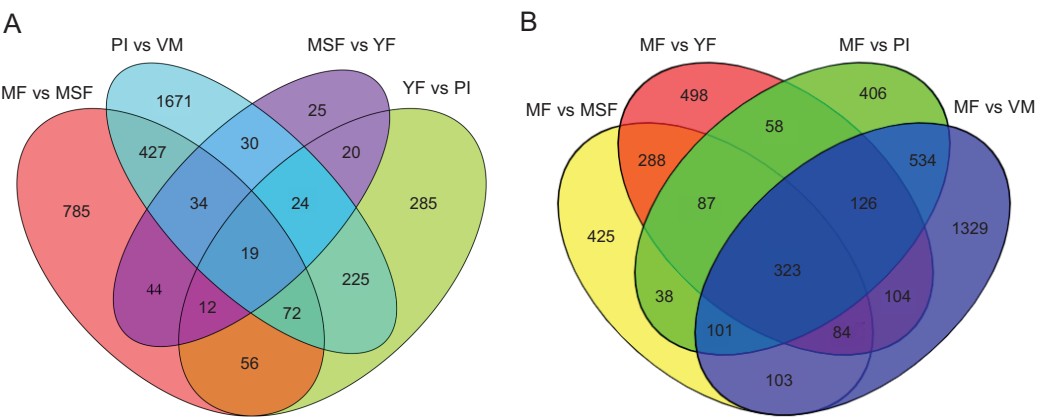

**Figure 5 Venn diagrams of different pairwise comparisons were introduced to show the statistics of the differentially expressed genes in more detail.** Venn diagrams showing the intersection of significant DEGs between themycelium, primordium and fruiting bodies in *L. decastes*. (A) Comparison between adjacent developmental periods. (B) Comparison between mature fruiting body and other developmental periods.

**Table 1 Some main up-expressed and down-expressed DEGs were detected between different development stages of *L. decastes*.** Statistics of the main up-expressed or down-expressed unigenes.

| Gene ID | Annotation | Log$_2$ fold change | | | |
|---|---|---|---|---|---|
| | | PI *vs* VM | YF *vs* PI | MSF *vs* YF | MF *vs* MSF |
| TRINITY_DN24293_c10_g1_i5 | Hypothetical protein | 9.90 | – | – | 2.03 |
| TRINITY_DN23461_c2_g1_i4 | Putative transport protein | 10.06 | −6.11 | – | – |
| TRINITY_DN23721_c7_g1_i7 | Guanine nucleotide-binding protein | 8.54 | −4.73 | – | – |
| TRINITY_DN21081_c2_g1_i1 | Hydrophobin | 7.73 | 1.35 | – | 1.05 |
| TRINITY_DN16793_c0_g3_i1 | Unknown protein | 6.90 | −3.08 | – | – |
| TRINITY_DN23419_c5_g1_i6 | Formamidase | 6.78 | −5.06 | – | – |
| TRINITY_DN9395_c2_g11_i1 | Unknown protein | 6.73 | – | −1.51 | – |
| TRINITY_DN3094_c0_g2_i1 | Phosphate dikinase | 6.31 | – | – | 2.56 |
| TRINITY_DN21184_c1_g4_i2 | Hypothetical protein | −5.99 | – | – | −1.20 |
| TRINITY_DN24419_c1_g7_i3 | Histone deacetylase | −5.98 | – | – | 1.20 |
| TRINITY_DN24528_c6_g1_i16 | Hypothetical protein | −6.11 | – | 3.87 | – |
| TRINITY_DN23783_c1_g4_i1 | Topoisomerase 1-associated factor | −6.73 | 3.04 | – | – |
| TRINITY_DN23086_c2_g1_i30 | Hypothetical protein | −7.33 | – | – | 1.40 |
| TRINITY_DN15818_c1_g1_i1 | Unknown protein | −8.67 | −1.85 | – | – |
| TRINITY_DN20519_c0_g1_i2 | Hypothetical protein | −9.18 | −1.39 | – | – |
| TRINITY_DN21533_c1_g2_i4 | Hypothetical protein | −9.42 | – | – | 2.34 |

DEGs in each subcategory was determined using the GO database, and the count ranged from 1 to 329 with Log$_{10}$ *p*-vlaue > 1. The significance threshold for the analysis was set at a corrected *p*-value of ≤0.05. For different comparison groups, the DEGs were significantly enriched in 26, 23, 20, 23, 25, 26 and 27 GO categories in PI *vs* VM, YF *vs* PI, MSF *vs* YF, MF *vs* MSF, MF *vs* YF, MF *vs* PI, and MF *vs* VM, respectively (Figs. 6 and S6). After

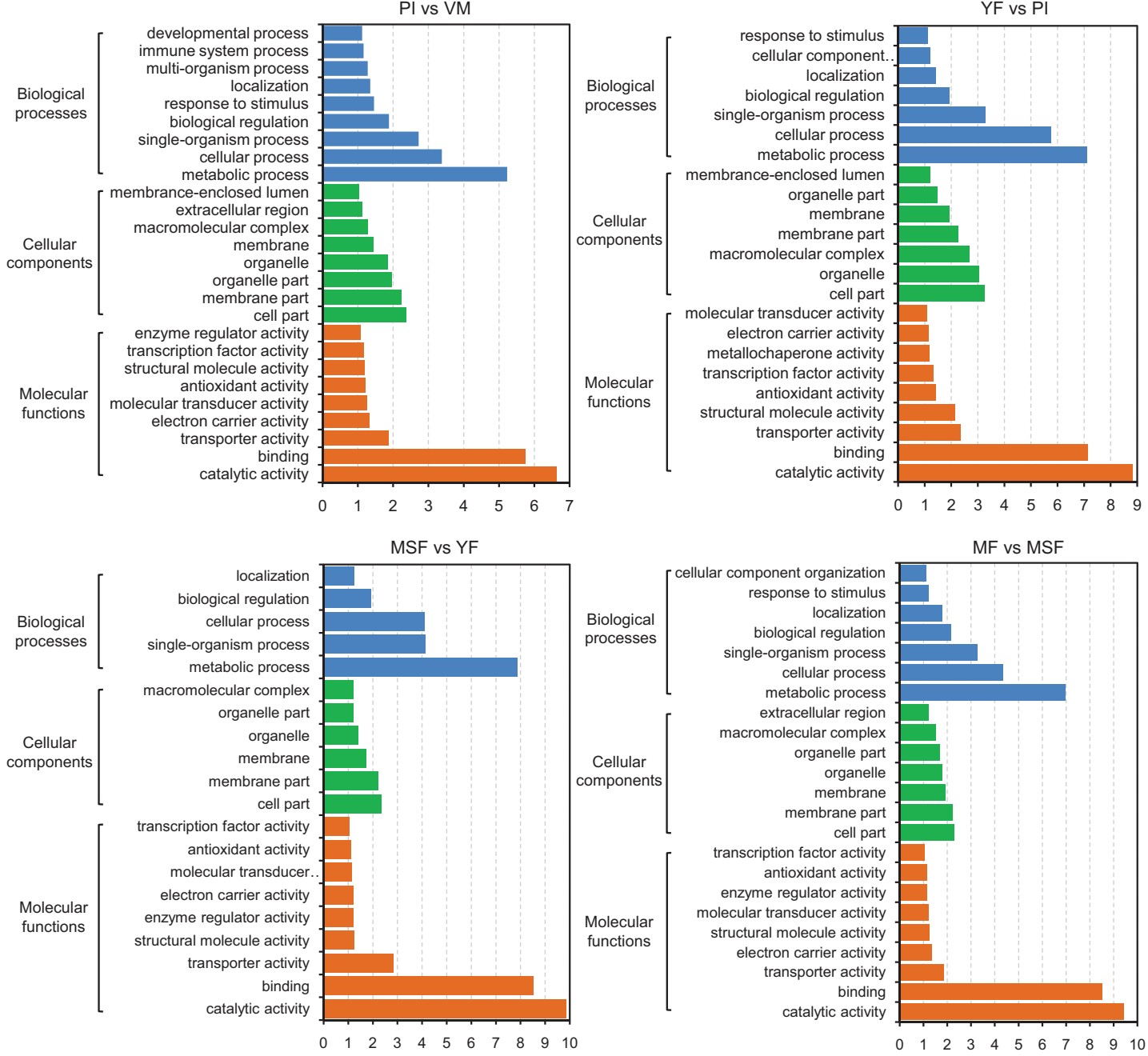

**Figure 6 The identified DEGs could be classified into three primary GO categories which contained 30 functional subcategories, with a range of one to 329 DEGs in each subcategory.** Gene ontology (GO) classification analysis of DEGs in different comparisons (PI *vs* VM, YF *vs* PI, MSF *vs* YF, and MF *vs* MSF). The x-axis indicates the enrichment score (−log$_{10}$ p-value) of GO terms enriched among DEGs. The y-axis indicates the specific GO term, and the orange bars represent molecular functions, the green represent cellular components, the blue represent biological processes. One unigene could be assigned to more than one category.

analyzing the seven comparison groups, a significant enrichment of the categories "metabolic process" (GO:0008152), "cell part" (GO:0044464), and "catalytic activity" (GO:0003824) within the domains of biological process, cellular component, and molecular function, respectively, was observed. Furthermore, a substantial proportion of DEGs were

linked to functional categories such as "cellular process" (GO:0009987), "single-organism process" (GO:0044699), "membrane part" (GO:0044425), and "binding" (GO:0000988). A certain number of DEGs belonged to the biological process category "response to stimulus" (GO:0050896) and molecular function category "antioxidant activity" (GO:0016209) in different comparison groups except for comparison MSF *vs* YF comparison. Analogously, no DEGs in MSF *vs* YF and YF *vs* PI were annotated to molecular function categories "developmental process" (GO:0003006), multi-organism process (GO:0051704) and "immune system process" (GO:0002376).

Our research focused on investigating the molecular mechanisms involved in the formation and development of fruiting bodies. To accomplish this, we conducted a GO functional classification analysis of DEGs at various stages of development. Our analysis revealed that the up-regulated DEGs in the PI *vs* VM and MF *vs* VM comparisons were primarily associated with the GO categories of "cellular process" (GO:0009987), "single-organism process" (GO:0044699), "metabolic process" (GO:0008152), and "binding" (GO:0000988). Conversely, there were relatively few down-regulated DEGs in these comparisons. However, in other comparisons, the opposite trend was observed (Figs. 6 and S6). These findings suggest that the expression profiles of genes associated with these GO categories may differ between the mycelium and fruiting body stages of *L. decastes* development.

## Pathway enrichment analysis of DEGs

To further scrutinize the biological functions of the DEGs and synthesize the biological pathways that pertain to the fruiting process, we mapped the DEGs from each comparison to the KEGG database (Figs. 7, S7 and Table S7). Interestingly, in the five developmental stages of *L. decastes*, the DEGs were only enriched in five pathways in YF *vs* PI, and there were 13, 17, and 16 pathways in PI *vs* VM, MSF *vs* YF and MF *vs* MSF, respectively. The KEGG sub-classification analysis revealed that a significant number of DEGs were implicated in metabolic pathways (ko01100), ribosome (ko03010), spliceosome (ko03040), biosynthesis of second metabolites (ko01110) and RNA transport (ko03013). These pathways are associated with genetic information processing and metabolism, and their prominence in the comparison between PI and VM indicates that they may play a pivotal role in the conversion of mycelium to primordium. Nevertheless, we found no DEGs enriched in the pathway of glycerolipid metabolism (ko00561) in the comparison of PI *vs* VM, but found them in the rest of the comparisons. In addition, there were 16, 15 and 13 DEGs assigned to the pathway glycosphingolipid biosynthesis-globo (ko00603) in PI *vs* VM, YF *vs* PI and MSF *vs* YF, respectively, but not in MF *vs* MSF. The comparative analysis of MSF *vs* YF, MF *vs* MSF, MF *vs* YF and MF *vs*, VM revealed the presence of DEGs that were enriched in several pathways related to environmental information processing. These pathways included the MAPK signaling pathway (ko04013), FoxO signaling pathway (ko04068), HIF-1 signaling pathway (ko04066), TNF signaling pathway (ko04668), and VEGF signaling pathway (ko04370). These results suggest that various signaling pathways may play crucial roles in the transition from a young fruiting body to mature fruiting body.

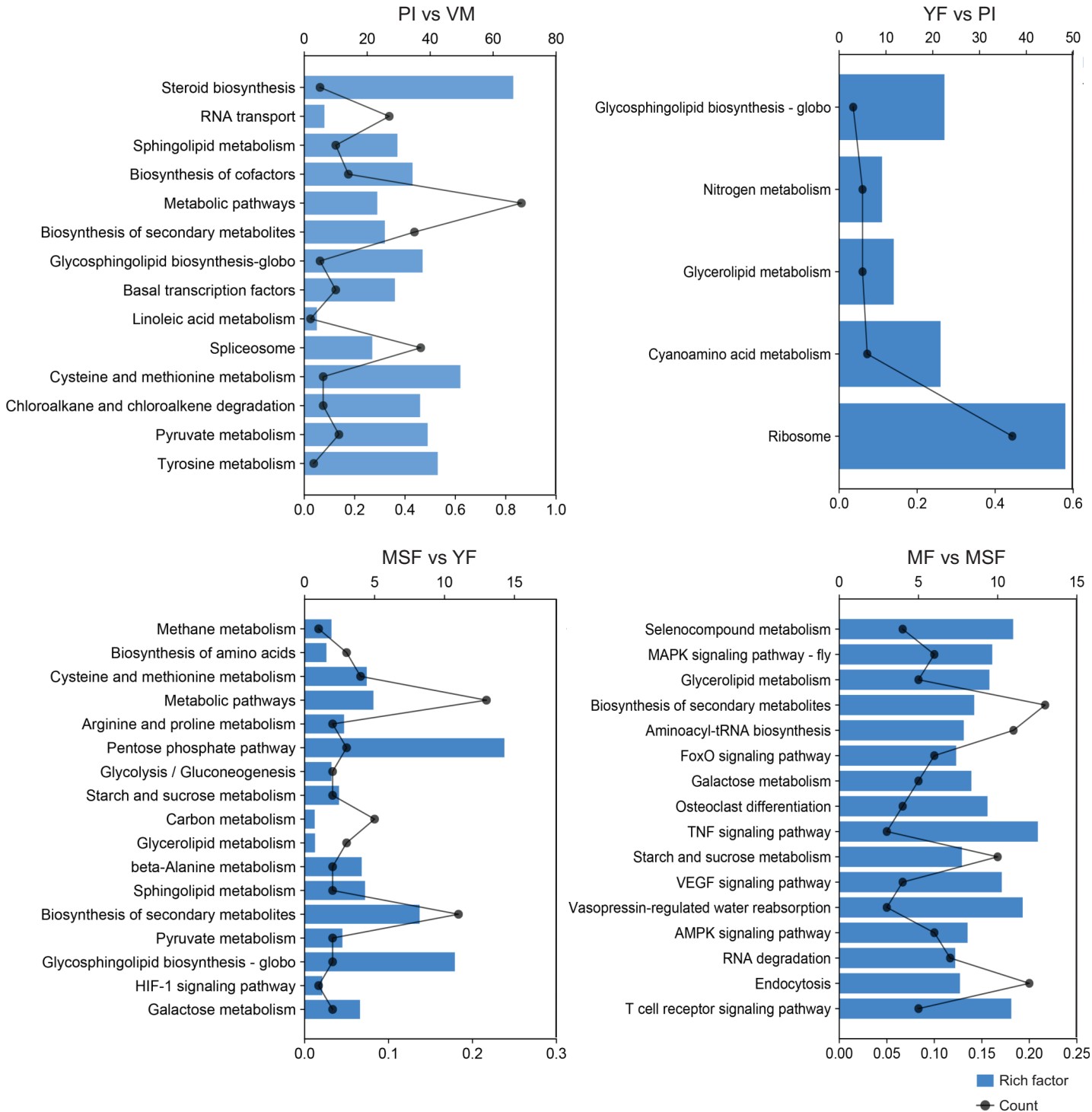

**Figure 7 In order to further evaluate the biological functions of the DEGs and summarize the biological pathways related to the fruiting process, the DEGs in each comparison were matched to the KEGG database.** The KEGG pathway enrichment analysis of DEGs in different comparisons (PI *vs* VM, YF *vs* PI, MSF *vs* YF and MF *vs* MSF). The *x*-axis indicates the rich factor. The *y*-axis indicates the pathway. The size of the dots indicates the number of DEGs in the corresponding pathway.

Meanwhile, we focused on the specific DEGs that were enriched in different pathways across multiple comparisons. We found that TRINITY_DN24453_c5_g1_i2, a cell cycle checkpoint control protein galactosidase, was simultaneously enriched to 14 metabolic pathways of PI *vs* VM, YF *vs* PI, MSF *vs* YF and MF *vs* MSF. Similarly, three other DEGs, TRINITY_DN21188_c3_g1 (hypothetical protein), TRINITY_DN23599_c5_g3 (polyketide synthase) and TRINITY_DN24271_c1_g2 (hypothetical protein) were simultaneously identified in the various pathways of MF *vs* MSF, MF *vs* YF, MF *vs* PI and MF *vs* VM.

## Special genes involved in fruiting body development

Previous investigations found that the analysis of DEGs offered valuable information regarding the development of the fruiting body. This process is initiated by a stimulus that activates particular genes responsible for the transformation of mycelium into a fruiting body (*Yu et al., 2012*; *Muraguchi et al., 2015*; *Song, Kim & Kim, 2018*; *Yan et al., 2023*). To identify the specific unigenes that contribute to the development of fruiting bodies in *L. decastes*, we conducted an analysis of the gene expression profiles across different comparisons. Our results indicated that 41 extracellular enzyme genes exhibited significant differences in expression levels at various stages, including eight cellulase genes, 11 galactosidase genes, 14 laccase genes, and eight peroxidase genes (as outlined in Table 2). Among these DEGs, there were 29, 14, three and 24 in PI *vs* VM, YF *vs* PI, MSF *vs* YF and MF *vs* MSF, respectively. The statistics also showed that a total of five, six and six extracellular enzyme genes were exclusively differentially expressed in PI *vs* VM, YF *vs* PI and MF *vs* MSF, respectively. In particular, the unigene TRINITY_DN24453_c5_g1_i2 that encoded a galactosidase was found to be significantly down-regulated in PI *vs* VM and MSF *vs* YF and up-regulated in YF *vs* PI and MF *vs* MSF. However, none of the laccase and peroxidase genes were differentially expressed in MSF *vs* YF or MF *vs* MSF.

Additionally, our primary emphasis was directed toward DEGs that fell under the category of transcription factor families. As a result, we systematically compiled a list of 19 potential transcription factors that were categorized into 14 distinct types. All of these types exhibit noteworthy variations in their expression levels throughout diverse developmental stages of *L. decastes*, as demonstrated in Table 3. Among of them, different amounts of several types transcription factors were characterized, including bHLH, C2H2-type, MADS box, MYB, and TFIIA, *etc*. While significant changes in expression were observed in these transcription factor genes, their patterns of up-regulation or down-regulation varied across different comparisons. For instance, the MYB gene TRINITY_DN24473_c6_g6_i2, was down-regulated in PI *vs* VM and MSF *vs* YF, up-regulated in YF *vs* PI and not detected in MF *vs* MSF. In summary, the transcription factor genes manifested dynamic expression profiles while the fruiting body of *L. decastes* was forming.

Our analysis led us to conclude that the extracellular enzyme genes and transcription factor genes found in *L. decastes* played a role in the initiation and maturation of fruiting bodies. Moreover, the discovery and examination of the distinctive genes of *L. decastes*

**Table 2 Transcriptome data analysis, we found that many extracellular enzyme-related genes showed significant differential expression during the fruiting body development of *L. decastes*.** Statistics of significant differently expressed extracellular enzyme genes.

| Gene ID | Annotation | Log₂ fold change | | | |
|---|---|---|---|---|---|
| | | PI *vs* VM | YF *vs* PI | MSF *vs* YF | MF *vs* MSF |
| TRINITY_DN24324_c8_g2_i1 | Cellulase | −2.29 | −3.41 | −1.08 | − |
| TRINITY_DN22377_c1_g2_i13 | Cellulase | 1.41 | 1.61 | − | − |
| TRINITY_DN22671_c3_g4_i1 | Cellulase | −2.22 | − | 1.46 | − |
| TRINITY_DN22138_c3_g5_i1 | Cellulase | −1.31 | − | − | −2.58 |
| TRINITY_DN21757_c0_g1_i1 | Cellulase | 1.01 | − | − | −2.04 |
| TRINITY_DN23506_c4_g1_i6 | Cellulase | 5.36 | −6.14 | − | −5.07 |
| TRINITY_DN23926_c6_g1_i8 | Cellulase | −3.63 | − | − | 2.19 |
| TRINITY_DN20438_c0_g1_i1 | Cellulase | 1.41 | 1.85 | − | − |
| TRINITY_DN23177_c0_g2_i18 | Galactosidase | 2.51 | | | 2.95 |
| TRINITY_DN24453_c5_g1_i2 | Galactosidase | 8.53 | −3.46 | 3.4 | 1.68 |
| TRINITY_DN21048_c4_g1_i1 | Galactosidase | −5.06 | −2.13 | − | − |
| TRINITY_DN8603_c0_g5_i1 | Galactosidase | − | −2.39 | − | − |
| TRINITY_DN23474_c0_g1_i3 | Galactosidase | − | 1.43 | − | − |
| TRINITY_DN10844_c0_g1_i1 | Galactosidase | 3.57 | −2.08 | − | − |
| TRINITY_DN17886_c0_g2_i1 | Galactosidase | − | −1.83 | − | − |
| TRINITY_DN29525_c0_g1_i1 | Galactosidase | −5.72 | − | − | 1.10 |
| TRINITY_DN23493_c0_g1_i1 | Galactosidase | − | − | − | 1.33 |
| TRINITY_DN24190_c2_g4_i2 | Galactosidase | − | − | − | 1.07 |
| TRINITY_DN23601_c4_g1 | Galactosidase | 10.43 | − | − | −6.39 |
| TRINITY_DN22183_c0_g1_i6 | Laccase | 7.04 | − | − | 3.19 |
| TRINITY_DN24358_c3_g5_i10 | Laccase | 8.19 | − | − | − |
| TRINITY_DN23702_c1_g1_i2 | Laccase | −2.27 | − | − | −1.26 |
| TRINITY_DN23325_c2_g2_i7 | Laccase | −2.00 | − | − | − |
| TRINITY_DN24358_c3_g5_i14 | Laccase | −2.13 | − | − | − |
| TRINITY_DN22840_c2_g2_i1 | Laccase | −1.66 | − | − | 1.60 |
| TRINITY_DN23468_c6_g6_i3 | Laccase | −3.56 | − | − | − |
| TRINITY_DN23011_c3_g1_i6 | Laccase | 6.50 | 1.49 | − | 6.00 |
| TRINITY_DN24413_c5_g4_i1 | Laccase | − | 1.30 | − | − |
| TRINITY_DN20512_c1_g2_i3 | Laccase | −5.35 | − | − | 2.08 |
| TRINITY_DN22996_c0_g1_i4 | Laccase | −1.20 | −4.68 | − | −5.90 |
| TRINITY_DN22840_c2_g1_i5 | Laccase | − | − | − | 2.01 |
| TRINITY_DN23702_c0_g6_i1 | Laccase | − | − | − | −1.12 |
| TRINITY_DN24413_c4_g2_i1 | Laccase | 5.37 | − | − | − |
| TRINITY_DN24411_c5_g1_i1 | Peroxidase | 4.38 | −3.62 | − | − |
| TRINITY_DN21840_c2_g4_i3 | Peroxidase | 1.45 | −3.04 | − | 1.95 |
| TRINITY_DN23165_c2_g1_i1 | Peroxidase | − | − | − | 1.96 |
| TRINITY_DN23224_c1_g2_i11 | Peroxidase | −3.19 | − | − | 3.26 |
| TRINITY_DN4946_c0_g1_i1 | Peroxidase | − | 1.56 | − | − |
| TRINITY_DN21580_c2_g3_i2 | Peroxidase | − | −1.23 | − | −2.05 |
| TRINITY_DN21929_c1_g1_i3 | Peroxidase | −4.01 | − | − | −2.50 |
| TRINITY_DN24087_c3_g2_i5 | Peroxidase | − | − | − | 2.30 |

Table 3 **Different types of transcription factors that might play important roles in fruiting body formation, such as C2H2 type zinc finger, GATA zinc finger, HMG-box, MADS-box, and TFIIA.** Statistics of significant differently expressed transcription factors.

| Gene ID | Type | Log$_2$ fold change | | | |
|---|---|---|---|---|---|
| | | PI *vs* VM | YF *vs* PI | MSF *vs* YF | MF *vs* MSF |
| TRINITY_DN21641_c1_g1_i2 | C2H2-type | – | −2.5 | 1.05 | −1.17 |
| TRINITY_DN23823_c0_g1_i12 | C2H2-type | −1.06 | – | – | −1.3 |
| TRINITY_DN23871_c0_g1_i5 | GATA-4/5/6 | −6.57 | – | −1.54 | – |
| TRINITY_DN23609_c4_g2_i1 | GATA-4/5/6 | −1.74 | – | – | −1.14 |
| TRINITY_DN24245_c2_g1_i1 | HAP1 | 2.81 | 1.23 | – | – |
| TRINITY_DN24594_c6_g5_i5 | HOX domain | 1.16 | – | 1.18 | −1.71 |
| TRINITY_DN21686_c0_g1_i1 | IIB | 2.07 | 2.98 | – | – |
| TRINITY_DN23623_c0_g3_i3 | IIIA | −1.36 | – | – | −1.31 |
| TRINITY_DN23813_c1_g2_i7 | JmjC domain | −1.34 | – | – | −1.55 |
| TRINITY_DN23787_c0_g6_i2 | MADS box | 1.31 | −1.27 | – | – |
| TRINITY_DN23117_c1_g6_i1 | MADS box | −3.49 | – | – | 4.27 |
| TRINITY_DN24473_c6_g2_i2 | MYB | −1.65 | 1.55 | −1.62 | – |
| TRINITY_DN23886_c2_g1_i1 | NirA | 2.1 | 1.22 | – | – |
| TRINITY_DN22724_c3_g1_i9 | Prr1 | – | 1.05 | – | −1.3 |
| TRINITY_DN23461_c2_g1_i4 | SFP1 | 10.06 | −6.11 | – | – |
| TRINITY_DN24454_c7_g1_i1 | TFIIA | 1.82 | – | – | −1.53 |
| TRINITY_DN23842_c3_g11_i2 | TFIIA | −5.45 | −1.21 | 2.2 | −2.9 |
| TRINITY_DN21757_c0_g1_i1 | TFIIA | 1.01 | – | – | −2.04 |
| TRINITY_DN24290_c5_g2_i3 | TFIIH | – | −1.31 | – | −1.42 |

provide significant genetic knowledge for future investigations into the molecular processes involved in the growth of fruiting bodies.

## Validation of RNA-Seq data by qRT-PCR

In order to validate the transcriptome data, we analyzed specific genes encoding extracellular enzymes that were associated with the development of fruiting bodies using qRT-PCR to determine their expression patterns. The results were in agreement with those obtained by RNA-Seq. The conformity between the two approaches affirmed the trustworthiness of our RNA-Seq data (Fig. 8).

## Analysis of extracellular enzyme activity

By releasing extracellular enzymes, fungi can effectively transform complex macromolecules into smaller units, thereby providing the necessary nutrients for their growth and development. Through the above transcriptome data analysis, we found that many extracellular enzyme-related genes showed significant differential expression during the fruiting body development of *L. decastes* (Table 2 and Fig. 8). To investigate the alterations in extracellular enzyme activity of *L. decastes* during various stages of growth and development, we established a theoretical framework to comprehend the utilization

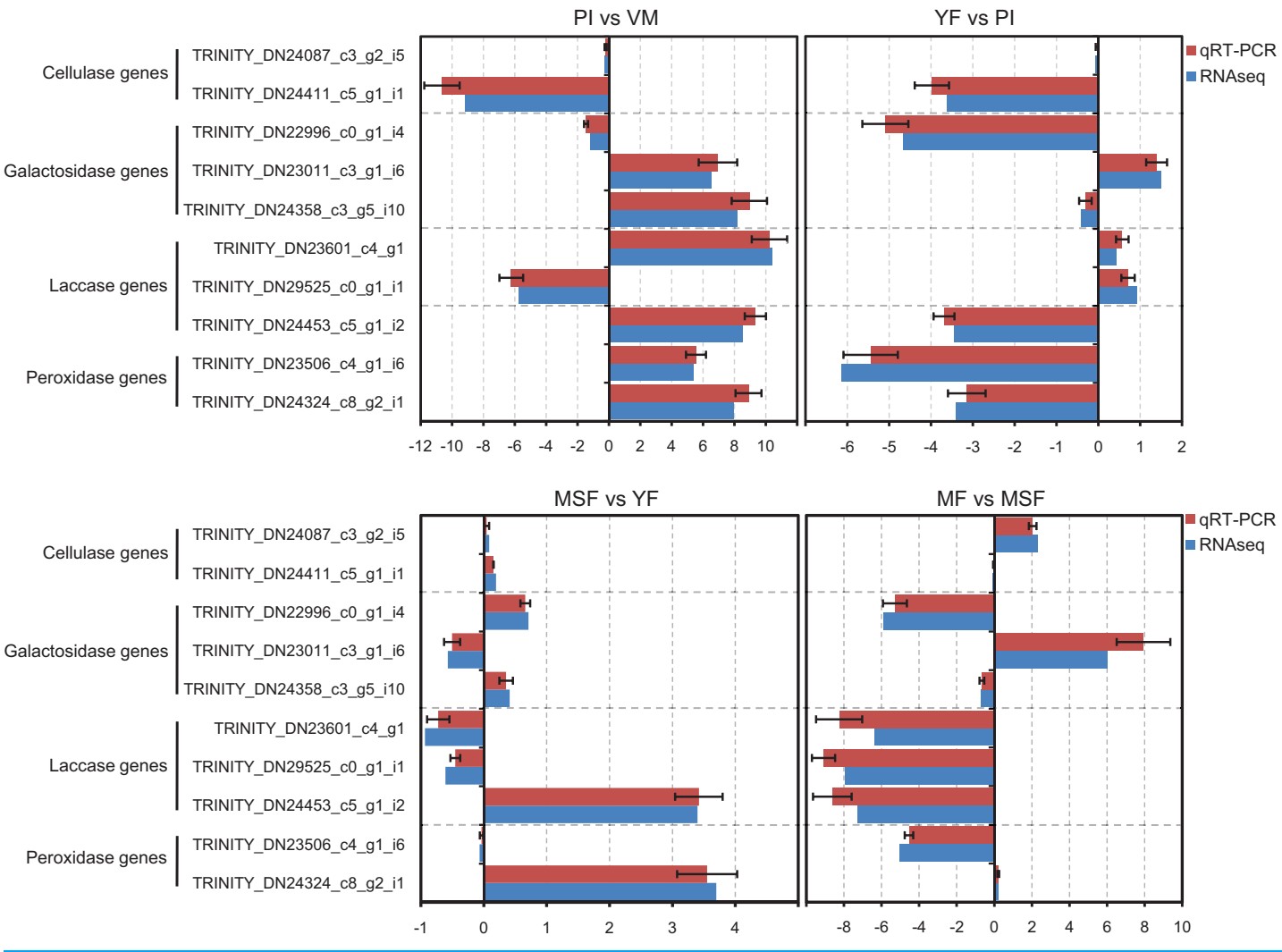

**Figure 8** **The expression profiles obtained by RNA-seq were validated by qRT-PCR analyses for selected several extracellular enzymes genes, which were related to fruiting body development.** qRT-PCR validation of the relative expression data of 12 genes involved in fruiting body development obtained in RNA-Seq analysis. The *x*-axis represent the fold change at different fruiting body development stages in *L. decastes*.

mechanism of *L. decastes* in a medium. We analyzed the activities of cellulase, galactosidase, laccase and peroxidase across five developmental stages (VM, PI, YF, MSF, and MF) of *L. decastes*. The experimental results showed that all four extracellular enzymes showed high activity in primordial initiation stage, and then decreased gradually (Fig. 9). The activities of cellulase, galactosidase and laccase all reached the maximum values at the primordium initiation stage, while the activities of peroxidase reached the maximum value at the vegetative mycelium stage (Fig. 9). The enzyme activity experiments conducted on *L. decastes* demonstrated that the extracellular enzymes exhibited discernible differences across various stages of growth and development. In order to elucidate and confirm the differences of these extracellular enzymes in different stages, expression levels of three cellulase regulators were selected and analyzed by qRT-PCR (Fig. S8). We found that the expression level of transcript inhibitors *CRE1* and *ACEI* was significantly down-regulated

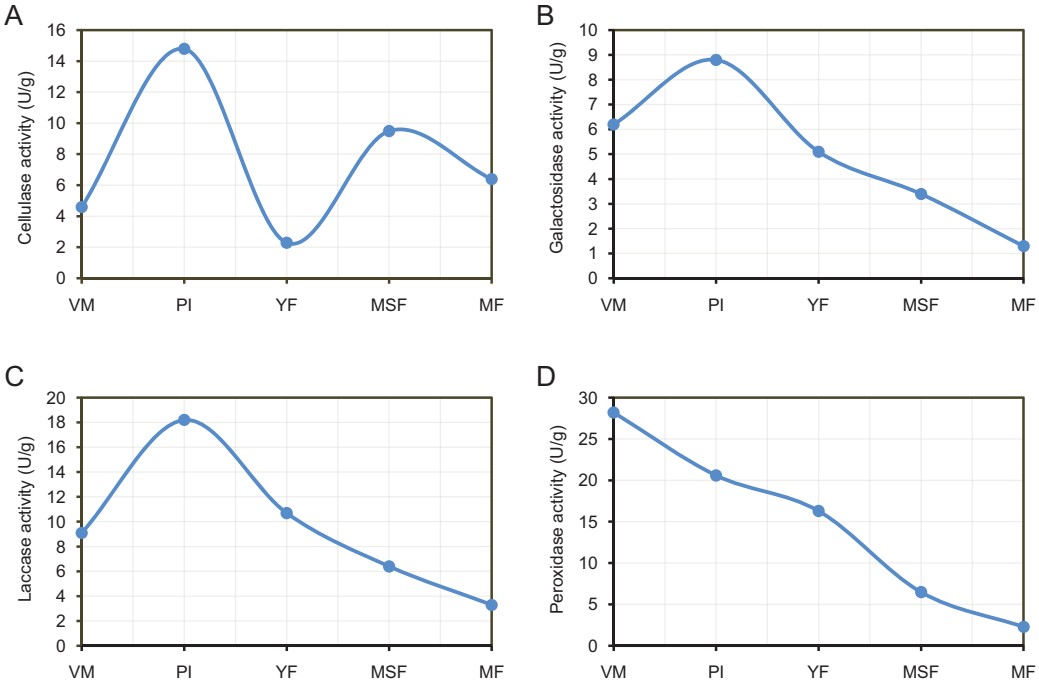

**Figure 9 The experimental results show that all four extracellular enzymes showed high activity in the primordial initiation stage, and then decreased gradually.** Trend of enzyme activities of extracellularenzyme in *L. decastes* during different growth and development periods. (A) Cellulase activity. (B) Galactosidase activity. (C) Laccase activity. (D) Peroxidase activity.

at the primordium initiation stage, and then obviously up-regulated at the young fruiting body stage. The expression difference in the patterns of transcription activator *ACEII* is similar to that of cellulase activity between different developmental stages. The subsequent examination of enzyme activity provided essential insights into the pivotal role that extracellular enzymes play in the growth and development of the fruiting body in *L. decastes*.

# DISCUSSION

## Extracellular enzyme-related genes are important for the formation of *L. decastes* fruiting body

In edible fungi, the substrates used for cultivation are usually composed of wheat bran, cottonseed husk, sawdust, and other substances that possess significant nutrient molecules, including lignin, cellulose, hemicelluloses, starch, and protein (*Huang et al., 2019*). To meet the demands of growth and metabolism, edible fungi are stimulated to produce hemicellulose, cellulose, and various extracellular hydrolases during their development. These enzymes are essential for the degradation, assimilation, conversion, and utilization of large nutrient molecules (*Lechner & Papinutti, 2006*; *Rani, Kalyani & Prathiba, 2008*; *Barh et al., 2022*; *Guigón-López et al., 2014*). For example, *Volvariella volvacea* produces multiple forms of extracellular laccase when grown in specific culture conditions, and some laccase genes may play an important role in the development of fruiting bodies (*Chen*

*et al., 2003*; *Lu et al., 2015*). Studies have found that amplifying the operation of laccase stimulates the initiation of *H. marmoreus* primordia and elevates the production of fruiting bodies (*Zhang et al., 2015b*; *Zhang et al., 2016*). Organisms' carbohydrate metabolismrely heavily on lignocellulolytic enzymes, which are crucial for the transformation of the lignocellulolytic matrix during fungal growth and serve as a necessary physiological function (*Elisashvili et al., 2002*; *Bánfi et al., 2015*). The peak of extracellular cellulase activity was positively correlated with the biomass of fruiting bodies in *Agaricus bisporus* (*Claydon, Allan & Wood, 1988*). Moreover, the importance of peroxidase in the growth of mycelium and the development of fruiting bodies in *H. marmoreus*, *Botrytis cinerea*, and *Ganoderma lucidum* has been demonstrated (*Kim et al., 2011*; *Ren et al., 2017*; *Chen et al., 2018*).

In the current study, a group of genes encoding extracellular enzymes such as cellulose, galactosidase, laccase and peroxidase were up-regulated or down-regulated from mycelium to mature fruiting body stages (Table 2). The analysis of qPCR and enzymeactivityalso indicated that the gene expression and enzyme activity of extracellular enzymes showed significant differences at different developmental stages (Figs. 8 and 9). Surprisingly, the differences in enzymatic activity and the expression levels showed different patterns of change, such as galactosidase activity and galactosidase genes (Figs. 8 and 9). We hypothesize that the process of extracellular enzymes affecting fruiting body development is controlled by a complex regulatory network involving multiple genes, and futher work is need to characterize the mechanism of these extracellular enzymes and related genes in fruiting body development. Specifically, the gene encoding galactosidase (TRINITY_DN24453_c5_g1_i2) was not only significantly differentially expressed in each comparison but was categorized into 14 KEGG pathways. The outcomes of this investigation are consistent with earlier studies, indicating that the aforementioned genes are likely linked to the regulation of fruiting body formation in *L. decastes* through direct or indirect involvement, but further research is needed. Therefore, additional investigation is essential in order to recognize and evaluate the existence and roles of these extracellular enzyme genes in the growth of *L. decastes* mushrooms.

## Transcription factors are essential for the formation of *L. decastes* fruiting body

Transcription factors act as molecular switches, binding to specific DNA sequences to regulate gene expression, and participating in many important cellular processes. In fungi, transcription factors that regulate fruiting body formation have been identified (*Ohm et al., 2011*; *Park, 2013*; *Xiang et al., 2014*; *Hamann, Osiewacz & Teichert, 2022*). In this research, we identified some different types of transcription factors that might play important roles in fruiting body formation, such as C2H2 type zinc finger, GATA zinc finger, HMG-box, MADS-box and TFIIA (Table 3). In filamentous fungi, C2H2 zinc finger protein is critical for the regulation of carbon metabolism and the expression of cellulase and hemicelluloses which are related to cellulose degradation, such as *Trichoderma reesei* cellulase regulators CRE1, ACE, and *Neurospora crassa* 11 hemicellulose regulators XLR-1(*Martinez et al., 2008*; *Sun et al., 2012*; *Campos Antonieto et al., 2014*). It has been found that *exp1*, a

HMG-box transcription factor is a crucial factor in fruiting body morphogenesis and takes part in the regulation of fruiting body opening and autolysis (*Muraguchi et al., 2008*). Previous research has determined that there are two *L. edodes* transcription factors, *PriA* and *PriB*, that are involved in the modulation of gene expression during the process of primordium formation (*Endo et al., 1994*; *Miyazaki et al., 2004*). In addition, during the development of fruiting bodies in *S. commune*, there was an up-regulation of *fst4*, a transcription factor that is specific to fungi (*Ohm et al., 2011*). Recently some transcription factors including MADS, C2H2, and FST4 were identified in the mushroom formation of *P. tuoliensis* (*Fu et al., 2017*). Therefore, it is postulated that the transcriptional regulators coded by these unigenes are noteworthy genetic resources for forthcoming studies on the inception and maturation of primordia, along with the formation of fruiting bodies in *L. decastes*. However, it needs more data and experiment areas to prove it.

## Signaling pathways in *L. decastes* fruiting body development

KEGG pathway analysis is a useful way to explore the metabolic and regulatory networks in organisms and is often used to search for pathways with significant enrichment in DEGs. According to earlier investigations, a significant number of KEGG pathways were pertinent to the formation of fruiting bodies in fungi, which included primary carbohydrate metabolism and the MAPK signaling pathway (*Ceccaroli et al., 2011*; *Fu et al., 2017*). The KEGG pathway analysis of different developmental stages of *L. decastes* revealed that a significant number of DEGs were enriched in various signaling pathways, as illustrated in Figs. 7 and S7. This observation is consistent with previous studies on other species, highlighting the correlation between signaling pathways and fungal fruiting body development. For instance, *IDC2* and *IDC3* genes were involved in signaling pathways and linked to *Podospora anserine* fruiting body development (*Lalucque et al., 2017*). Similarly, the MAPK signaling pathway was found to be up-regulated during fruiting body development in *Coprinopsis cinerea* (*Cheng et al., 2013*). However, studies about MAPK signaling *L. decastes* in were lacking. It should be emphasized that the DEGs involved in signaling pathways were exclusively associated with processing environmental information. This implies that environmental cues may significantly impact the development of fruiting body morphology. These findings collectively indicate that signaling pathways are essential for *L. decastes* fruiting body formation. The elucidation of the molecular mechanisms involved in *L. decastes* fruiting body development necessitates further investigation into the DEGs mentioned above. Consider that the current work involved only the transcriptome data, it is only possible to predict some possible functions of identified DEGs, and their specific regulatory mechanisms and networks need to be further studied and clarified.

## CONCLUSION

A comprehensive comparative analysis of the transcriptome of *L. decastes* was conducted using RNA sequencing across five consecutive developmental stages. The method employed yielded a considerable number of transcripts that were subsequently analyzed to explore the biological mechanisms implicated in the maturation of fruiting bodies.

Functional enrichment analysis identified the genes and pathways linked to the transitions between various developmental stages. This was accomplished by utilizing GO and KEGG classification of DEGs. Moreover, the examination detected modifications in the expression levels and enzyme activity of particular extracellular enzymes during the five developmental periods. The aforementioned findings supply valuable information into the molecular mechanisms that underlie the development of fruiting bodies in *L. decastes*, as well as precise genetic data that can be utilized as targets for molecular breeding and further research. Our study provides valuable genetic information for further studies of molecular mechanism aiming to improve productivity and quality, and establish a developmental trajectory of *L. decastes* in future.

## ACKNOWLEDGEMENTS

We thank all the participants for their generous contributions.

### Funding

This work was supported by the Science and Technology Project of Gansu Province, China (NOs. 21YF5NA127, 23YFNG0006, 23YFNG0004), the National Natural Science Foundation of China (NO. 31860582), the Longyuan Young Innovative and Entrepreneurial Talents (Team) Project, China (NO. 2022LQTD14) and the Doctoral Research Foundation of Hexi University (NO. KYQD2020060). The funders had no role in study design, data collection and analysis, decision to publish, or preparation of the manuscript.

### Grant Disclosures

The following grant information was disclosed by the authors:
Science and Technology Project of Gansu Province, China: 21YF5NA127, 23YFNG0006 and 23YFNG0004.
National Natural Science Foundation of China: 31860582.
Longyuan Young Innovative and Entrepreneurial Talents (Team) Project, China: 2022LQTD14.
Doctoral Research Foundation of Hexi University: KYQD2020060.

### Competing Interests

The authors declare that they have no competing interests.

### Author Contributions

- Shanwen Ke conceived and designed the experiments, performed the experiments, prepared figures and/or tables, and approved the final draft.
- LingQiang Ding performed the experiments, analyzed the data, prepared figures and/or tables, and approved the final draft.
- Xin Niu performed the experiments, prepared figures and/or tables, and approved the final draft.

- Huajia Shan analyzed the data, prepared figures and/or tables, and approved the final draft.
- Liru Song analyzed the data, prepared figures and/or tables, and approved the final draft.
- Yali Xi analyzed the data, authored or reviewed drafts of the article, and approved the final draft.
- Jiuhai Feng analyzed the data, authored or reviewed drafts of the article, and approved the final draft.
- Shenglong Wei analyzed the data, authored or reviewed drafts of the article, and approved the final draft.
- Qianqian Liang conceived and designed the experiments, authored or reviewed drafts of the article, and approved the final draft.

## Data Availability

The transcriptome sequencing raw data are available at the National Center for Biotechnology Information (NCBI) Sequence Read Archive (SRA): PRJNA962251.

## Supplemental Information

Supplemental information for this article can be found online at http://dx.doi.org/10.7717/peerj.16288#supplemental-information.

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
