# Peer review of "Comparative transcriptome analysis on candidate genes associated with fruiting body growth and development in Lyophyllum decastes"

_PeerJ, doi:10.7717/peerj.16288_

## Round 0.1 · original submission · Major Revisions

Four experts in this field assessed your manuscript and thought that after revision might be suitable for publication in this journal. Please address all comments and concerns. In particular, those related to data analysis, and results description.

·

Basic reporting

Ke and colleagues report here the transcriptome analysis of Lyophyllum decastes , described an RNA-seq analysis of the different developmental stages of Lyophyllum decastes. The manuscript is well written, and the majority of the data presented is relevant to the manuscript. This reviewer has some questions regarding the manuscript I explain in the following lines. All of my comments are in goodwill, hoping to clarify some issues detected.
As a general suggestion, the presentation of the results could be improved, specifically the reference to Figures; this is inconsistent and should be sequential. One figure is not cited in the text. Also, the manuscript could benefit by including a model as a final figure. In the next sections, I provide more specific items.

Experimental design

Overall, the experimental design I sound, the methods section is sufficiently described, the bioinformatics is robust, and the authors use well-established and validated tools. However, this reviewer detected some issues (see 'Validity of the findings section' for comments) that are associated with the curation and analysis; here this reviewer is a bit confused because, in the methods section and experimental design, I cannot see why there is an issue regarding the unigene numbers, please see the next section.

Validity of the findings

Although the manuscript contains relevant and insightful information, I kindly request the authors to provide the melt curves of all qRT-PCR experiments as supplementary material to strengthen the confirmation of DEGs in this study. This is relevant to show that the primers and amplification products are unique and no issues are detected in this type of experiment.
In the results section, the authors state that the assembly suggests over 76,000 unigenes, however, in doi.org/10.3389/fmicb.2023.1137162, the total number of predicted genes in the genome of this organism is 14,000. Also, the total amount of data may be overestimated; perhaps authors pooled the data. Here, I suggest checking if this was the case to report the number of unigenes per experiment and separately report vs. unique sequences on each dataset. Also, please verify the length of the contigs (line 178). I think this number is incorrect (37.46 bp); this number must be bigger. Also, please provide here the average FastQC values. My concern with the number of unigenes is that only 43,000 were annotated. Perhaps the assembly generated hybrid sequences; please check; for this reviewer, this is a major concern. Another solution that I can think of is to perform a second assembly. Still, this time use as reference the genomes sequenced and compare how good the de novo assembly is compared with this reference genome assembly. Also, Figure 2 is problematic; here, the vast majority of the unigenes are in the range of 200 to 500 bp. For this reviewer, this could be related to deficient cDNA synthesis.
Overall, the whole results section regarding the annotation and analysis needs to be carefully revised and be consistent within this manuscript and now with the available genome sequence (doi.org/10.3389/fmicb.2023.1137162, GenBank accession numbers CP107701-CP107715 and JAOWSY000000000 for strain LRG-d1-1 and LRG-d1-5, taken from the same reference). Therefore, the results section should be revised, too, to ensure that the data is correctly presented, specifically for the DEGs in each developmental stage.
The GO analysis requires a better presentation; I think authors would greatly benefit by ordering the GO terms by function and then presenting the GO terms with the expression ratio in a single figure; perhaps using a heat map could be more illustrative. In this way, in the same figure, we compare GO terms present on each data set comparison and the expression ratio.
In lines 283 and 284, the authors found a KEGG sub-classification and found photosynthesis and carbon fixation pathways. Please address this in the context of the reported genome; are genes related to photosynthesis in the genome? If this is the case, there is a breakthrough in fungal biology, and these genes should be analyzed in detail. However, why did the authors that reported the genome sequence miss these genes? I kindly request to clarify this issue and if this is a misinterpretation on my side.
In lines 321 to 338, the authors mention a set of transcription factors that led to conclude that the extracellular enzymes and transcription factors are involved in the initiation and maturation of fruiting bodies; in this case, I strongly recommend including a table of the role of the transcription factors, based on their homology and compare this information with other fungal organisms and correlate the relevance of the pathway found in this work.
In this reviewer's opinion, the statement in lines 340 to 344 and Fig.8 and Table S7 will be clearer if, in a single figure (perhaps in Fig. 8), the data from the RNA-seq experiment is included for the same genes. In order to make this comparison, I suggest using fold induction or fold change; in this way, the fold change with each technique on top of each bar can be compared.
I kindly request to discuss the differences in enzymatic activity (Figure 9) and the expression levels shown in Fig. 8; I can see differences, and perhaps this is more related to the way is presented in the text, I apologize if I am missing something here.
In the case of Table 3, I find that in some instances, the fold change is not that great. How did authors decide to include transcription factors in this analysis when they saw a 1-fold expression change in only one comparison? I kindly suggest discussing this in the rebuttal letter and pondering whether it is more informative to maintain the discussion only with transcription factors showing a bigger expression difference or at least in two or more differentiation stages. I think that the manuscript could benefit from assembling a story where the extracellular enzymes and their regulators are the main story; in that sense, I recommend determining by qRT-PCR the expression levels of some regulators, such as CRE1, and now having a complete story with the expression levels and enzymatic activity, which in turn may be more stable in this organism and accumulate extracellularly and thus explaining the differences with the expression pattern. In the same line of thought, I also recommend including a model showing the pathway known to morphological differentiation in fungi (known to date), including the transcription factors found here and how well the model fits with other edible fungi, and discussing differences.
Overall, the conclusions of the work agree with the manuscript's content, and the results support the finding of genes associated with the development of Lyophyllum decastes.

Additional comments

In line 51, this reviewer thinks that it should be modified to "the fruiting body formation is regulated at the transcriptional level" Although this line seems incomplete, I guess the authors needed to add something like "environmental signal regulates at the transcriptional level, the star of the morphological changes…" or something in the like.
In line 57, I suggest modifying to Polysaccharides.
In lines 100-101, do the authors mean "in the dark"? I think this is better.
Why do authors use diverse RNA purification kits/methods in the methods section? They should explain this further in lines 110-112 if this is relevant. If this is not relevant, please remove it.
In the methods section should be referenced the figure 1. This figure is nowhere referenced. I think this figure is relevant since it may help non-experts to know the morphology analyzed here.
Please correct in line 181; the average size authors reported in Table S2 is 1124.61 bp.
Table 1 title, I suggest using "up or down-regulated"
In the results section, I recommend using a different nomenclature when referring to TRINITY unigenes, perhaps using the annotation of the homologs.

Reviewer 2 ·

Basic reporting

The authors conducted a comprehensive literature review on Lyophyllum decastes and current existing comparative transcriptomic studies. One suggestion is to include numbers in the abstract (e.g., how many differentially expressed genes are identified).

Experimental design

In this manuscript, Ke et al. used RNA-seq to study the development of fruiting body of L. descastes. The article is well written and here’re some minor suggestions:

• Please include the references of the databases and specify software version used in this article. For example, several databases (e.g., Nr, GO, COG, KEGG, and Swiss-Prot) are not properly cited in the “Assembly and functional annotation” section.
• In the “Expression and enrichment analysis” section, the DESeq and multiple testing correction were introduced. Also, p-value cutoff was used to identify significant GO terms. However, the authors used p-value<0.05 as cutoff to determine, which might be a typo. Please clarify.
• In the “Expression and enrichment analysis” section, please briefly describe the steps of KEGG pathway of the in-house scripts in words.
• In the results – Fig4, please include a volcano plot to highlight specific genes that has been discussed in each comparison.
• Is it possible to use transcriptome to establish a developmental trajectory/pseudotime?
• Is it possible to assign a time (e.g. day 14, 28, etc) to each developmental stage? If so, it will provide a continuum of developmental trajectory and can analyze the DEG/pathway in a continuous fashion.
• Please share the code and in-house scripts (e.g., github) and specify the parameters/version of the software tools to ensure reproducibility.

Validity of the findings

no comment.

·

Basic reporting

The authors have made commendable efforts to address numerous research questions using various methodologies and extensive data analysis. In order to improve the clarity of reporting, I would like to suggest the following recommendations:

1) The language used in the paper tends to be technical, which may hinder understanding for readers who are not experts in the field. It would be beneficial to make the language more accessible by avoiding excessive jargon and providing full explanations or acronyms for terms, such as RIN and FPKM, especially when they are first introduced in the paper.

2) Potential typo error in line 238. Please review and correct any such errors to maintain accuracy of the manuscript.

3) In line 253, incorrect GO categories numbers are mentioned for some of the comparison groups. It is recommended to update the correct numbers from Figure 6 and Fig S5 to ensure accuracy and consistency in reporting.

4) In Figure 6, what does the green color indicate - please specify.

5) To enhance the credibility of the information provided, it would be beneficial to include specific table and figure references where necessary. For example, in line 282-283, it is mentioned that the KEGG sub-classification analysis revealed significant numbers of DEGs implicated in specific pathways. Providing a reference to support this claim will strengthen the validity of the statement.

By implementing these recommendations, the paper will become more reader-friendly, accurate, and well-supported, ultimately enhancing its overall quality.

Experimental design

No comments

Validity of the findings

The authors aimed to gain a deeper understanding of the molecular mechanisms involved in the development of the fruiting body of Lyophyllum decastes, a highly regarded mushroom with culinary and medicinal properties. They thoroughly described the methodology used, supported by data. This study provides valuable insights into the molecular mechanisms underlying the development of the fruiting body in L. decastes.

However, there are a few areas where the authors could provide more details:

1) Providing a clearer context: It would be beneficial to explain in more detail why understanding fruiting body development is important. This could include discussing the potential impact of this research on mushroom yield, quality improvement, and the development of new medicinal applications. By highlighting these aspects, readers will better understand the significance of the study.

2) Addressing potential limitations: It would be helpful if the authors acknowledged any limitations of their study. This could include discussing factors that may have influenced the results or areas where further research is needed to fully understand the molecular mechanisms involved in fruiting body development. Acknowledging limitations will provide a more balanced perspective and help readers interpret the findings appropriately.

3) Elaborating on future research directions: The conclusion briefly mentions the potential use of the findings for molecular breeding and further research. It would be beneficial to expand on the specific areas of future investigation, such as targeted gene manipulation or functional studies. This would provide a roadmap for researchers interested in building upon this work and exploring specific aspects of fruiting body development in L. decastes.

Reviewer 4 ·

Basic reporting

The objective of the study seems to be appealing and presented extensively. Context and background is clear. Figures could however be with better resolution.

Experimental design

While experimental design seems to be solid I would suggest authors to go through the method section again and mention version of the Bioconductor package or any software used in the analysis for example line 133 and 143. Some point though-

1. I also wonder why authors did not use the DESeq2 and used DESeq instead. Any explanantion regarding that?

2. The authors talk about pre-processing of the paired end RNA-Seq data but did not show any plots regarding quality of the raw data .

3. How would authors justify only a small matching of unigenes matched in all databases. Is it something that authors expected?

Validity of the findings

A few points regarding results-

1. Kindly go through all typos and punctuation again . Figure legend also needs to be punctuated properly.

2. When authors mention about up or downregulated genes they should also clearly mention what specific stage they are indicating to. I understand it can be deciphered looking at comparison set but it is always clearer if mentioned in text also.

3. Figure 6 is very confusing . The legend does not match the plots and very misleading. Needs to be reviewed in both text and plot.

4.Line 325-327 is also not well elaborated. It should be re-written clearly.

5.Figure 5- I am not sure if 28 genes found DE in adjacent developmental periods and 907 genes found in comparison be maturing body and other stages are mutually exclusive.

6. Authors should also include p-values in table 3

Additional comments

None

---

## Round 0.2 · Minor Revisions

Three experts assessed again your manuscript and think this version has addressed the main concerns previously raised. However, two Reviewers are still pointing out areas where the manuscript can be improved. Please address these comments in a new version of the manuscript.

·

Basic reporting

This reviewer has no more comments on this section.

Experimental design

This reviewer finds that the manuscript in this section has been improved greatly.

Validity of the findings

For this reviewer, the manuscript improved greatly. I only have two concerns. First, the melt curve for TRINITY_DN4946_c0_g1_i1 and TRINITY_DN22377_c1_g2_i13 either should be eliminated form the paper or must be repeated. The most worrisome melt is for DN4946_c0_g1_i1,, that is providing a serious defective melt, specifically for W0.
In the rebuttal letter, point 8 response to reviewer 1, I think that authors are just eliminating the finding of sets of genes that may have been obtained either by soil contamination or that for some reason this fungus obtained by HGT. I find a bit odd that authors simply eliminated this from the manuscript. I was expecting a more robust response on this regard, specifically, to revise the raw data and check for possible plant DNA contamination. Please address this point in more depth.
The rest of the manuscript is now acceptable for publication.

Additional comments

In rebuttal letter point 13 for reviewer 1, I think that authors if are unable to provide a detailed model, at least they can provide a relative time-to-growth gene expression pattern discussion. This is not mandatory but I think will wrap nicely the results section.

Reviewer 2 ·

Basic reporting

The authors have addressed all comments. No further suggestions.

Experimental design

The authors have addressed all comments. No further suggestions.

Validity of the findings

The authors have addressed all comments. No further suggestions.

·

Basic reporting

No comments

Experimental design

No comments

Validity of the findings

No comments

Additional comments

The authors have made significant enhancements to the manuscript, taking into account the recommendations from all four reviewers. They have been commendably receptive to acknowledging errors, demonstrating a willingness to incorporate changes. Their efforts to provide additional data to further refine the manuscript are particularly noteworthy. Their patience in addressing all queries from the reviewers has been appreciable. I advocate for the manuscript's approval, contingent upon the completion of a few minor revisions.

1) Please ensure correct spacing for lines 153-158.
2) Lines 176-177: The statement "The fragments per kilobase of exon model mapped fragments (FPKM) values for these genes are available" appears to lack a reference.

---

## Round 0.3 · accepted · Accept

The authors addressed properly all the Reviewers' concerns. As a consequence, this work is ready for publication.